# Cardiotoxicity of Novel Targeted Hematological Therapies

**DOI:** 10.3390/life10120344

**Published:** 2020-12-11

**Authors:** Valentina Giudice, Carmine Vecchione, Carmine Selleri

**Affiliations:** 1Department of Medicine, Surgery and Dentistry “Scuola Medica Salernitana”, University of Salerno, Baronissi, 84081 Salerno, Italy; cvecchione@unisa.it (C.V.); cselleri@unisa.it (C.S.); 2Clinical Pharmacology, University Hospital “San Giovanni di Dio e Ruggi D’Aragona”, 84131 Salerno, Italy; 3IRCCS Neuromed (Mediterranean Neurological Institute), 86077 Pozzilli, Italy; 4Hematology and Transplant Center, University Hospital “San Giovanni di Dio e Ruggi D’Aragona”, 84131 Salerno, Italy

**Keywords:** cardiotoxicity, targeted therapy, adverse reactions, hematology

## Abstract

Chemotherapy-related cardiac dysfunction, also known as cardiotoxicity, is a group of drug-related adverse events negatively affecting myocardial structure and functions in patients who received chemotherapy for cancer treatment. Clinical manifestations can vary from life-threatening arrythmias to chronic conditions, such as heart failure or hypertension, which dramatically reduce quality of life of cancer survivors. Standard chemotherapy exerts its toxic effect mainly by inducing oxidative stress and genomic instability, while new targeted therapies work by interfering with signaling pathways important not only in cancer cells but also in myocytes. For example, Bruton’s tyrosine kinase (BTK) inhibitors interfere with class I phosphoinositide 3-kinase isoforms involved in cardiac hypertrophy, contractility, and regulation of various channel forming proteins; thus, off-target effects of BTK inhibitors are associated with increased frequency of arrhythmias, such as atrial fibrillation, compared to standard chemotherapy. In this review, we summarize current knowledge of cardiotoxic effects of targeted therapies used in hematology.

## 1. Introduction

In last decades, scientific advances in onco-hematology have significantly improved outcomes of cancer patients who have become long-term survivors; however, they must face late and long-term treatment-related effects that worsen their quality of life [1]. In particular, chemotherapy-related cardiac adverse events, also known as cardiotoxicity, represent the most common cause of death in long-term survivors—children after 15–25 years of cancer diagnosis have a cumulative incidence of heart failure (HF) of 4.4% and a rate of cardiac death 8.2-fold higher than the age- and sex-matched general population [2,3]. Myocardial disfunction and HF are the most common and life-threatening manifestations. Cardiotoxicity can also display as coronary artery disease (CAD), valvular diseases, arrhythmias, peripheral vascular disease (PAD) or stroke, arterial hypertension, and any other cardiovascular manifestations, such as pulmonary hypertension [1,4,5,6]. Cardiotoxicity can be evident right after the first dose exposure to several years after the end of treatment depending on patient’s characteristics, disease biology, type of chemotherapy administered, and cumulative dose [1]. Therefore, patients should be regularly monitored to early identify and treat cardiotoxicity, and echocardiography is the most used, simple, and widely accessible method for monitoring cardiac function and vascular system [1,7]. For example, echocardiography can be used for early detection of chemotherapy-related cardiac dysfunction (CTRCD), defined as a persistent reduction of the left ventricular ejection fraction (LVEF) >10% compared to baseline levels and confirmed 2 to 3 weeks after the first measurement [1,8]. Prolonged augmented circulating troponin levels might also help in early detection of left ventricular (LV) dysfunction during chemotherapy treatment, while (N-terminal pro) brain natriuretic peptide (BNP/NT-proBNP) levels can increase after therapy [9].

Cardiotoxicity caused by “old” standard chemotherapeutic drugs, such as alkylating agents and anthracyclines, or first-generation targeted therapies, such as tyrosine kinase inhibitors (TKIs), is well-known and characterized. Chemotherapy is employed for treatment of Hodgkin and non-Hodgkin lymphomas, myeloid and lymphoid leukemias, and before hematopoietic stem cell transplantation as conditioning regimens. Chemotherapy-related cardiac complications and mortality have been historically reported at high incidence rates in earlier studies; however, optimization of cumulative dosage, such as introduction of reduced-intensity conditioning regimens, or different chemotherapeutic scheme, such as from combination of vincristine, methotrexate, cyclophosphamide, and prednisone (MOMP) to adriamycin, bleomycin, vinblastine, and dacarbazine (ABVD) for treatment of Hodgkin lymphoma, has significantly reduced the impact of cardiotoxicity in hematological patients [10,11]. In the last decades, standard chemotherapy has also been associated or replaced with novel targeted therapies that are used for rescue refractory/relapsed patients. Cardiac adverse events of novel targeted therapies are still under investigation because of the short available follow-up, especially for recently approved drugs [1]. In this review, we provide an update of cardiotoxic effects of novel targeted therapies used in malignant hematological disorders with a particular focus on TKIs, proteasome inhibitors, immunomodulatory drugs (IMiDs), and demethylating agents.

## 2. TKIs

TKIs are a group of drugs that selectively inhibit tyrosine kinases, enzymes responsible of signal transduction after receptor stimulation by adding a phosphate group from ATP to a tyrosine residue on proteins [12]. In several hematological disorders, tyrosine kinases are constitutionally activated, leading to a deregulation of signaling pathways involved in cell survival and proliferation [13]. For example, in chronic myeloid leukemia (CML), a reciprocal translocation (9;22)(q34;q11), known as “Philadelphia chromosome” (Ph), determines the formation of a fusion protein, B cell receptor (BCR)/ABL, that activates various intracellular pathways, such as signal transducer and activator of transcription 5 (STAT5) and phosphoinositide 3-kinase (PI3K) signaling pathways, leading to increased cell survival and proliferation. Therefore, by blocking this kinase with specific inhibitors, neoplastic clone growth and survival is halted, favoring apoptosis and cell death and thus reducing tumor burden [12].

### 2.1. Bruton’s Tyrosine Kinase Inhibitors

Bruton’s tyrosine kinase (BTK), a non-receptor member of TEC kinase family, is essential for B cell development as mutations in BTK cause a primary immunodeficiency X-linked agammaglobulinemia [14]. BTK is important in downstream of B cell receptor (BCR) and chemokine signaling cascades, and in cell survival because BCR engagement increases anti-apoptotic protein expression and S-phase cyclins [15,16,17]. In addition, BTK plays a crucial role in cell adhesion signaling pathways, including integrin α4β1 (VLA-4) and vascular cell adhesion molecule-1 (VCAM-1)/fibronectin pathways [18]. In particular, VLA-4 is involved in bone marrow (BM) homing and retention of hematopoietic cells because of its ability to interact with VCAM-1 on BM stromal cells [18,19,20]. Moreover, VLA-4 can interact with CD38 and form the macromolecular complex involved in trans endothelial rolling, invasion, arrest, and survival of the neoplastic clone in BM and lymphoid tissues [18,21]. These BTK functions are essential in survival and proliferation of malignant cells in various non-Hodgkin lymphomas (NHL), such as chronic lymphocytic leukemia (CLL) and mantle cell lymphoma (MCL) [14]. Therefore, BTK inhibitors can block tumor growth and induce apoptosis in neoplastic cells.

Ibrutinib, an oral irreversible BTK inhibitor, covalently binds the 481 cysteine of the kinase domain, blocking BTK activity but not interactions with Syk, and has been approved in 2016 for treatment of CLL, MCL, and Waldenström’s macroglobulinemia [14]. Despite its short life in clinical practice, ibrutinib is already known to cause cardiotoxicity, especially arrhythmias and hypertension, probably because of interactions with PI3K and other TEC pathways involved in cardiac protection under stress conditions (Figure 1) [22,23,24,25,26]. BTK inhibitors can interfere with all class I PI3K isoforms (PI3Kα, PI3Kα, PI3Kγ, and PI3Kδ), differently expressed in various tissues and involved in cardiac functions. Class I PI3Kα has an essential role in physiological cardiac hypertrophy and contractility, and is activated in cardiomyocytes by insulin or insulin-like growth factor-1 (IGF-1), while in endothelial cells, fibroblasts and vascular smooth muscle cells by fibroblast growth factor (FGF), platelet-derived growth factor (PDGF), and vascular endothelial growth factor (VEGF). In mouse models, PI3Kα suppression worsens hypertrophic cardiomyopathy caused by pressure overload or myocardial infarction (MI), while PI3Kα activation ameliorates hypertrophic and dilated cardiomyopathy [27,28]. PI3Kα is also involved in regulation of various channel-forming proteins, such as K^+^: Kir, Ca^2+^: Cav1, or Na^+^: SCN5A; direct inhibition of PI3Kα or at the receptor level (e.g., by ibrutinib) causes activation of late Na current (INa-L) through PIP3 reduction, resulting in enhanced action potential and QT prolongation. In addition, PI3Kα inhibition affects L-type Ca^2+^ current (ICa,L), and modulates Ca^2+^ cycling and α-adrenergic stimulation favoring action potential prolongation, abnormal automaticity, and early and delayed afterdepolarization [27]. While PI3Kγ, mainly expressed in leukocytes and cardiac cells, is upregulated during atherosclerosis, it has opposite inotropic functions compared to PI3Kα, and decreases myocardial β-adrenergic receptor (β-AR) under stress conditions, such as during congestive heart failure (CHF) [27,29]. Therefore, the use of agents that block the activation of INa-L (e.g., ranolazine) or upregulate ion channel expression might reduce the impact of PI3K inhibition-related cardiotoxicity [27,30].

The most common ibrutinib-associated arrhythmia is atrial fibrillation (AF), with an incidence of 5.77 per 100 person-years (PY) over a median follow-up of 18.3 months, significantly higher than that of general population [31,32]. Incidence of AF is increased in patients receiving ibrutinib compared to standard chemotherapy (6.5% vs. 1.6% over a 16.6 month follow-up), higher in combination with other drugs (7.7% vs. 5.8% of ibrutinib alone; HELIOS study) or longer follow-up (10.4% over a 78 month period). Median time to AF onset is 2.8 months, and incidence at 6 months is 5.3% [33]. Most patients develop de novo AF, and CLL subjects are more susceptible to this adverse event compared to MCL (7.0% vs. 4.3%). Other risk factors for ibrutinib-associated AF are prior history of AF, age >65 years, pre-existing hypertension and hyperlipidemia, and high Shanafelt risk score category in CLL. Prior CAD, valvular disease, and diabetes are not associated with increased risk of AF [33], while previous use of angiotensin-converting enzyme inhibitors (ACE-Is), angiotensin receptor blockers (ARBs), beta-blockers, and aspirin are associated with increased risk of AF in patients treated with ibrutinib [34]. Ventricular arrhythmias, such as QT/QTc prolongation, premature ventricular contractions (PVCs), non-sustained ventricular tachycardia (VT), ventricular fibrillation, and sudden cardiac death are also reported with a cumulative incidence of 1991 events per 100,000 PY [35,36,37].

Other forms of ibrutinib-associated cardiotoxicity are arterial hypertension, central nervous system (CNS) hemorrhagic or ischemic events, cardiomyopathy, and HF [38]. Arterial hypertension is a common cardiac adverse event during ibrutinib treatment, with a cumulative incidence of 78% and a variable median time to peak of blood pressure (BP) from 1.8 to 6 months; however, hypertension can develop after a very short time from initiation, and thus a close monitoring of patients in early months of ibrutinib treatment is required [39,40,41,42]. Newly diagnosed or worsening of a pre-existing hypertension during ibrutinib treatment is associated with a higher incidence of major adverse cardiovascular events, especially AF [39,40]. CNS hemorrhages or ischemia can frequently occur [38]. These contrasting events might be related to the ability of BTK inhibitors to variously interfere with platelet glycoprotein Ib (GPIb) and GPVI signaling pathways and to alter interactions with von Willebrand factor (VWF) [43]. Indeed, ibrutinib and second generation BTK inhibitor acalabrutinib might impair thrombus formation on atherosclerotic plaques and interfere with platelet functions and aggregation [34,43]. An increased bleeding risk is also related to pharmacologic interactions between ibrutinib and anticoagulants, such as apixaban, rivaroxaban, and dabigatran, metabolized by cytochrome CYP3A4 leading to augmented plasma concentrations [34].

### 2.2. PI3K Inhibitors

PI3K pathway plays an important role in B cell development, adhesion and migration, proliferation and survival, and immune functions [44]. Three PI3K inhibitors have been approved by the U.S. Food and Drug Administration (FDA) and the European Medicines Agency (EMA) for treatment of indolent NHL: idelalisib, copanlisib, and duvelisib [45]. Idelalisib is the first-in-class reversible, highly selective inhibitor of delta PI3K isoform, and the most common adverse events are diarrhea, pneumonia and pneumonitis, hepatotoxicity, thrombocytopenia, and skin rash [46,47]. Cardiovascular adverse events are not frequent; however, AF and peripheral edema can occur. No increase in cumulative incidence of pulmonary hypertension, QT prolongation, or PAD has been reported [48]. Idelalisib and duvelisib have a safer cardiovascular profile compared to copanlisib, a pan-class PI3K inhibitor of α and δ isoforms, which frequently cause infusion-related hyperglycemia and hypertension (57.1% and 54.8%, respectively), as well as diarrhea [49,50]. Hypertension usually develops within 2 h of the first cycle infusion with a mean systolic BP increase of 16.8 mmHg, and resolves within 24 h [51].

### 2.3. Isocitrate Dehydrogenase (IDH) Inhibitors

*IDH1* and *IDH2* catalyze oxidative decarboxylation of isocitrate to α-ketoglutarate (αKG) and CO_2_. Under physiological conditions, D-2-hydroxyglutarate (D2HG) is rapidly converted in αKG by an endogenous D2HG dehydrogenase enzyme [52]; when somatic mutations occur in *IDH1* and *IDH2*, mutant forms acquire a neo-morphic activity causing reduction of αKG to the oncometabolite R-2-hydroxyglutarate that competitively inhibits the endogenous D2HG dehydrogenase enzyme. The modifications lead to an intracellular accumulation of D2HG, epigenetic alterations, and impaired hematopoietic differentiation [52,53]. Ivosidenib, a mutant *IDH1* inhibitor, and enasidenib, a mutant *IDH2* inhibitor, have been approved for treatment of relapsed/refractory acute myeloid leukemia (AML). QT prolongation is a common cardiotoxicity during ivosidenib treatment with an incidence of 24.6% at starting dose of 500 mg daily, and 10.1% of those adverse events are of grade 3 or higher [53]. Recently, a case of myopericarditis and cardiogenic shock following an IDH inhibitor-induced differentiation syndrome (IDH-DS) has been reported during enasidenib treatment [54,55,56]. *IDH2* inhibitor can also cause QT prolongation [57].

### 2.4. Janus Kinase Inhibitor

Janus kinases (JAKs) are a family of tyrosine kinases widely involved in signaling transduction [58]. In myeloproliferative disorders (MPNs), increased activation of JAK/STAT pathways in hematopoietic stem cells (HSCs) causes uncontrolled proliferation and cytokine production [59,60]; however, hematopoiesis is not ineffective as in myelodysplastic syndromes (MDS), and patients show various grades of polycythemia and/or thrombocytosis, and extramedullary hematopoiesis with splenomegaly [60]. The three most common molecular alterations are a somatic G>T mutation in position 1849 of exon 14 of the Janus Kinase 2 (*JAK2*) with valine to phenylalanine substitution in codon 617 (V617F); a W>L/K/A substitution in exon 10, codon 10 of the thrombopoietin receptor (*MPL*) gene; and mutations in exon 9 of the calreticulin (*CALR*) gene as a 52-bp deletion (L367fs*46) or a 5-bp insertion (K385fs*47) [61].

In 2011, the first JAK1/2 inhibitor, ruxolitinib, was approved for treatment of primary and secondary myelofibrosis (COMFORT studies) [62], and in 2014 and 2019, also for treatment of polycythemia vera (PV) and acute graft versus host disease (GvHD) [63,64]. Ruxolitinib-associated cardiotoxicity is still not well characterized; however, arterial hypertension might be a frequent comorbidity, as systolic BP can significantly increase after 72 weeks of treatment without significant changes in diastolic BP [65]. Patients can experience a worsening of pre-existing hypertension or can either develop a new onset disease (from 64.7% at baseline to 69.1% after 72 weeks of treatment) [65]. Ruxolitinib can attenuate the effects of growth hormones on STAT5 phosphorylation and favor weight gain through inhibition of JAK/STAT signaling in adipose tissue contributing to hypertension development and probably to late-onset cardiovascular diseases (Figure 2) [65]. After 72 weeks of treatment, the proportion of obese patients can double, and about 21% of subjects moves up to a higher BMI class [65]. No electrocardiographic changes have been reported [66]. A case of pulmonary hypertension with left ventricular (LV) dysfunction after five years of ruxolitinib treatment has been recently reported [67].

Fedratinib, an oral, potent JAK2 inhibitor effective against wild-type and mutationally activated JAK2 and fms-like tyrosine kinase 3 (FLT3), was approved in 2019 for treatment of adult patients with intermediate-2 or high-risk primary or secondary myelofibrosis [68]. Cardiotoxicity of fedratinib is still under investigation; however, peripheral edema and HF are reported [69,70]. However, fedratinib has received a “black-box warning” because of the increased risk of fatal encephalopathy including Wernicke encephalopathy [71]. Other JAK inhibitors under investigation are momelotinib and the dual JAK/FLT3 inhibitor pacritinib that cause less hematological adverse events and cardiotoxicity [72].

### 2.5. BCR/ABL Inhibitors

All known BCR/ABL fusion protein variants are composed by an ABL tyrosine kinase domain constitutionally active downstream of various signaling pathways, such as PI3K and STAT, involved in gene expression, mRNA processing and maturation, and protein stability [73,74]. In HSC compartment, deregulation of these pathways translates in increased cell survival and proliferation, and impaired differentiation with suppression of granulocyte maturation [74]. Imatinib, the first-in-class oral TKI approved for treatment of CML in 2001, has pharmacological activity against ABL, BCR/ABL, platelet-derived growth factor receptor A (PDGFRA), and c-KIT on neoplastic cells, and also against ABL on normal cells [73]. Imatinib binds amino acid residues in the ATP binding site and stabilizes the inactive forms preventing autophosphorylation and thus switching off signaling transduction [74]. Because of its off-target effects, imatinib can cause different adverse events, including cardiotoxicity [75]. CML patients can develop CHF with New York Heart Association (NYHA) class 3–4 symptoms after a mean of 7.2 ± 5.4 months of treatment [75]. Those findings opened a controversial debate on imatinib safety and related cardiotoxicity, as discordant data have been reported during last decades, especially from small case series [76,77,78,79,80]. Although rare, CHF and LVEF depression might occur with an incidence of 0.7–1.8% after a long-course imatinib treatment (6 months or more) in older patients [80,81,82,83].

Dasatinib, a second-generation TKI, can similarly induce CHF, with a reported incidence of 2–4% [80,83,84,85]. Other types of cardiotoxicity are arrhythmias, asymptomatic QT prolongation, and pleural and pericardial effusion [80]. Dasatinib administration at 140 mg/daily has been associated with increased incidence of pleural effusion in up to 35% of cases, higher in those subjects with CML in accelerated and blast phase [86,87]. Pleural and pericardial effusion has been also reported at lower dose (100 mg or 50 mg daily) [88], while single daily dose administration might decrease pleural effusion rate within the first 12 months of treatment [89,90]. Risk factors are history of cardiac disease, hypertension, and use of dasatinib at twice-daily schedule [88].

Nilotinib, a second-generation TKI, is 30-fold more potent than imatinib in blocking BCR/ABL activity; however, this drug is one of the most cardiotoxic TKIs. The most frequent cardiac adverse event is dose-dependent QT prolongation and sudden cardiac death [83,84,91]. Nilotinib-associated arrhythmias are caused by off-target inhibitory effects on a potassium ion channel (Kv11.1) involved in delayed-rectifier K^+^ current in cardiac tissue, thus inducing QT wave alterations (Figure 3A) [91]. In addition, MI, acute coronary disease (ACS), and peripheral arterial occlusive disease (PAOD) are frequent, with incidence varying among studies [84]. Nilotinib has additional off-target effects on vascular tissue and pro-atherogenic activities causing arterial stenosis and vasospasm. Moreover, nilotinib induces metabolic modifications, such as increased cholesterol and glucose levels, contributing to increased cardiovascular risk [92,93], and has a direct cardiotoxic effect through caspase activation and apoptosis induction [94]. Median time to a cardiovascular event is 14.5 months (range, 2–68), and patients might experience recurrent diseases requiring several angioplasties and/or surgeries within a few months [95,96].

Bosutinib, a second-generation TKI active against SRC/ABL, has been approved for CML resistant or intolerant to prior TKI therapy [97]. Bosutinib-associated cardiotoxicity incidence is low (6.8%); however, refractory/relapsed CML patients who received bosutinib as second- or above-line therapy have an increased incidence of cardiac adverse events compared to those who received bosutinib as first-line therapy (7.7% vs. 4.8%, respectively), especially cerebrovascular events [97]. Among cardiotoxic manifestations, angina pectoris, CAD, and PAD are frequent (1.2%, 1.2%, and 2%, respectively), while aortic arteriosclerosis, peripheral coldness, venous insufficiency, deep vein thrombosis, Raynaud’s phenomenon, and PAOD are uncommon. Serious adverse events are more frequently represented by CAD and MI, while deaths are often caused by cerebrovascular accidents in younger patients, or cardiovascular events in older subjects. Patients with advanced disease or receiving bosutinib as second- or above-line therapy have a greater risk of serious events and death [97]. Hypertension can occur in 7.8% of cases, especially those with a history of hypertension; however, incidence is similar to that of CML subjects treated with other TKIs, such as imatinib [97]. Age ≥65 years is a risk factor of cardiotoxicity when bosutinib is administered as first- or above-line treatment, while Eastern Cooperative Oncology Group performance status (ECOG PS) >0, pre-existing cardiovascular disorders and/or diabetes, and history of hyperlipidemia/increased cholesterol are risk factors when bosutinib is administered as second- or above-line treatment [97]. Other reported cardiotoxicity is peripheral edema, CHF, AF, QT prolongation, and pericardial effusion [98].

Ponatinib, a third-generation TKI, shows activity against multiple kinases, such as SRC/ABL, fibroblast growth factor receptor 1 (FGFR1), PDGFRA, vascular endothelial growth factor receptor 2 (VEGFR2), c-KIT, and FLT3 in hematopoietic cells, as well as FGFR2/3/4 and RET in various tumors [99]. Ponatinib has been approved for CML and Ph-positive (Ph+) acute lymphoblastic leukemia (ALL) resistant or intolerant to prior TKI therapy or for patients carrying the point mutation threonine to isoleucine at codon 315 (T315I) in *BCR/ABL1* kinase domain, a frequent somatic mutation associated with resistance to first- and second-generation TKIs [100]. Despite its efficacy, ponatinib is one of the most cardiotoxic TKIs, causing CHF, arrhythmias, arterial occlusive events, and hypertension [101,102,103]. Mechanisms of cardiotoxicity are still under investigation; however, off-target effects, especially on PI3K and Akt signaling pathways, could induce cardiotoxicity (Figure 3B) [101]. Off-target FGFR inhibition causes modifications in in vitro proliferation and differentiation of cardiomyocytes; FLT3 and c-Jun blockade is related to apoptosis; while PDGFR, VEGFR, and c-Src inhibition induces contractile alterations [101]. In addition, ponatinib can have pro-atherogenic properties by promoting surface adhesion receptor expression, and by enhancing platelet activation and aggregation [101]. Cumulative incidence of CAD, PAD, and cerebrovascular events is 26% [101,104,105]. In chronic phase CML, ACS and MI are the most frequent manifestations (12% of cases) and can precede CHF with a median time to initial onset of 11.5 months; cerebrovascular and peripheral arterial occlusive events occur in 6% and 8% of cases, respectively, and venous thromboembolic events (VTEs) are reported in 5% of subjects [101,104,105]. Hypertension is also frequent (14%). Serious cardiac adverse events are represented by AF (6%) and angina pectoris (5%) [104]. Incidence of cardiotoxicity is related to dose intensity with the highest rate (42%) at 45 mg/daily. History of ischemia, age at study entry, baseline neutrophil count, and time to treatment are prognostic risk factors [103]. In addition, there might be a lag time between drug administration and onset of cardiovascular event, as 7% of patients experience cardiotoxicity after study discontinuation [103,104]. Cardiovascular events can occur in 7.1% of patients, cerebrovascular accidents in 3.6%, and peripheral vascular events in 4.9% of subjects, more frequently in patients with a history of cardiovascular diseases and/or the presence of one or more cardiovascular risk factors, such as hypertension, diabetes, hypercholesterolemia, and obesity [105]. Moreover, ponatinib-treated patients have an increased incidence of recurrent arterial occlusive events compared to those treated with dasatinib or bosutinib (76.7% vs. 64%, respectively) [106]. Risk stratification can be assessed using a clinical score (Systematic Coronary Risk Evaluation (SCORE)) based on sex, age, smoking status, systolic BP, and total cholesterol levels. Patients with SCORE >5% have a higher incidence of arterial occlusive events compared to those subjects with SCORE <5% (74.3% vs. 15.2%) [107]. Aspirin administration can lower cardiovascular risk, especially in patients aged >60 years [107].

### 2.6. Other TKIs

Gilteritinib is a novel FLT3 inhibitor approved in 2019 as a monotherapy for relapsed/refractory AML harboring *FLT3* mutations including internal tandem duplication (ITD), D835Y, or D835Y somatic mutations. FLT3 blockade induces apoptosis in leukemic cells that rely on FLT3 signaling pathway for survival [108]. The most frequent reported cardiotoxicity is peripheral edema (any grade, 24%), and QT prolongation (4.9%) requiring dose reduction; however, QTc interval >500 ms is uncommon (0.4%) [109]. Similarly, glasdegib, a selective oral inhibitor of Hedgehog signaling through Smoothened, can also rarely cause QT prolongation [110,111]. In contrast with gilteritinib-associated cardiotoxicity, midostaurin does not induce QT prolongation [112]. Midostaurin, a multi-target TKI, shows activity against protein kinase C (PKC), PDGFRα/β, cyclin-dependent kinase 1 (CDK1), SRC, Syk, c-KIT, VEGFR2, and FLT3 [102,113]. Midostaurin can block autophosphorylation of the endogenous wild-type *FLT3*, but also both ITD and D835Y forms [113,114]. Other reported cardiotoxicity is hypertension, pericardial effusion, and anecdotic interstitial lung injury with pulmonary hypertension [115].

## 3. Proteasome Inhibitors

Poly-ubiquitination is a tag-system used for identification of proteins that need to be degraded because they are redundant, misfolded, or unwanted, such as proteins involved in specific cell cycle phases [116]. Ubiquitinated proteins are then degraded by the 26S proteasome complex hyperactivated in multiple myeloma (MM) causing excessive intracellular removal of important proteins, such as tumor suppressor p53 and IκB (inhibitor of nuclear factor-κB (NF-κB)). This alteration in proteasome activity translates in the deregulation of several intracellular processes, including cell cycle and apoptosis control, pro-inflammatory cytokine signaling, and stress response, leading to increased survival and proliferation of neoplastic clones [116]. Therefore, specific inhibitors can block oncogenic progression by interfering with constitutive and immune proteasome activity. These two forms of proteasomes differ for the barrel-like structure composition of the 20S catalytic domain—the constitutive form is composed of β5, β1, and β2 subunits, while the immunoproteasome 20S core has the β5i, β1i, and β2i subunits, whose expression is regulated by pro-inflammatory cytokines [116].

Bortezomib, the first-in-class proteasome inhibitor, shows activity against the β5, β1, and β5i subunits, thus interfering with both constitutive and immune proteasome forms. Proteasome is important in protein homeostasis maintenance in tissues with high metabolic demand, such as cardiac and liver tissues, and inhibition might cause cardiomyocyte disfunction and HF [117]. In rats, bortezomib can induce left ventricular contractile dysfunction with mitochondrial modifications, and reduction in ATP synthesis and cardiomyocyte contractile functions [118]. In clinical trials, incidence of bortezomib-associated cardiotoxicity varies from 0% to 17.9%, with the highest incidence in elderly patients with MCL and in MM subjects receiving bortezomib as monotherapy; however, the risk of cardiotoxicity is not higher than that of patients treated with other chemotherapeutic drugs [119]. The most common cardiotoxicity is CHF (2–5%), particularly frequent in patients aged >70 years and after a median of 3.2 months after starting bortezomib [120]; cardiomyopathy; acute MI; and arrhythmias. Hospitalization rate for CHF is 5.76/100 PY, for acute MI is 2.57/100 PY, and for arrhythmias is 3.10/100 PY [121,122]. Complete heart block and acute left ventricular dysfunction have also been described in few case reports [123,124,125,126,127]. Arrhythmias are secondary to HF, and case reports show various manifestations, including AF, complete atrio-ventricular block (CAVB), premature atrial or ventricular complexes (PAC and PVC), sinus bradycardia or tachycardia (SB and ST), supraventricular or ventricular tachycardia (SVT and VT), and torsades de pointes [128].

Carfilzomib, a second-generation proteasome inhibitor, has a cardiotoxic activity similar to that of bortezomib, with carfilzomib-associated cardiac event rate of 27% vs. 16% of bortezomib-treated patients and similar cumulative incidences [126,129]. More than 7% of patients can experience CHF, pulmonary edema, or decreased ejection fraction with an overall mortality rate of 7% [130]. Hypertension is also frequent (14.3%), either new onset or worsening of pre-existing conditions. Cardiac adverse events usually occur after a median of 67.5 days of therapy and incidence rate remains similar during treatment duration [129,130]. The association of carfilzomib with dexamethasone does not increase frequency of hypertension and HF [131]. In addition, carfilzomib can frequently cause SVT; case reports have described SB, CAVB, PAC, AF, and sudden cardiac death [128]. Ixazomib, the first oral proteasome inhibitor, shows cardiotoxic effects not increasing in combination with lenalidomide; in particular, ixazomib can induce arrhythmias, hypertension, HF, and MI [116]. However, MM patients are frequently old and frail with high incidence of pre-existing cardiac and renal diseases or MM-associated comorbidities that negatively decrease cardiac functions and general status, which is a great bias when analyzing safety cardiovascular profile of novel drugs for MM treatment [123,132]. In addition, association of proteasome inhibitors with other cardiotoxic drugs, such as lenalidomide, which are the standard of care is another limitation in assessment of cardiac safety profile of this class of drugs. Finally, new therapies are investigated on heavily pre-treated patients, and, in the case of MM, previous treatments are highly cardiotoxic; therefore, a novel drug is often administered to a subject with an already reduced cardiac function or with other treatment-related cardiovascular diseases [123,126].

## 4. IMiDs

IMiDs are a milestone in MM treatment. Neoplastic plasma cells finely interact with the BM microenvironment to increase their survival and to escape from immunosurveillance. IMiDs can interfere with pro-survival and anti-apoptotic pathways in malignant plasma cells, and with microenvironment interactions [133,134]. Thalidomide, the first-in-class IMiD, has anti-angiogenic and anti-tumor necrosis factor (TNF) activities; the two synthetic analogues, lenalidomide and pomalidomide, also modulate T cell differentiation by increasing the frequency of central and effector memory CD8^+^ T cells, T regulatory cells (Tregs), natural killer (NK) and NKT cells, and myeloid derived suppressor cells, while decreasing effector terminally differentiated T cells and expression of co-inhibitory receptors [133,134].

The most common cardiotoxicity of IMiDs are arrhythmias, venous thromboembolism (VTE), myocardial infarction, and cerebrovascular events [132,135,136,137,138,139]. Thalidomide is frequently associated with SB and AF, while CAVB, SVT, PVC, VT, and sudden cardiac death are rarely reported [128]; lenalidomide can cause AF, while SB, SVT, VT, and sudden cardiac death are infrequent [128]; and pomalidomide is associated with AF [139]. Lenalidomide can cause new onset HF (1.22 events per 100 PY) or worsen a pre-existing condition (9.76 events per 100 PY) [137]. Results from the ASPIRE trial using lenalidomide plus dexamethasone with or without carfilzomib show that lenalidomide alone increases the incidence of hypertension, HF, ischemic HF, and dyspnea [132]. Arterial and venous thromboembolisms are frequent, and incidence increases when IMiDs are associated with proteasome inhibitors or other chemotherapeutic agents [139]. Thalidomide and lenalidomide alone are associated with thrombotic events in up to 26% of patients, while pomalidomide is associated with up to 5% [139,140,141]. Age and active uncontrolled disease are risk factors of VTE [135]; however, MM patients themselves have an increased risk of VTE compared to other cancer subjects because of the presence of a hypercoagulable status that leads to VTE in 10% of cases receiving standard chemotherapy [138]. Lenalidomide is also associated with increased risk of arterial thromboembolism with incidence of myocardial infarction of 1.98% and cerebrovascular accidents of 3.4% [138]. For this reason, lenalidomide has a “black box warning” for arterial thromboembolism events.

## 5. Demethylating Agents

Modifications in DNA methylation are frequent in solid tumors and hematological diseases and are related to aberrations in gene expression and genomic instability, leading to increased gene mutations and chromosomal abnormalities. Azacytidine and decitabine are the two demethylating agents approved for the treatment of MDS and AML because of their ability to inhibit DNA methylation and induce cell differentiation, thus reducing ineffective hemopoiesis occurring during MDS and AML [142]. Phase II clinical trials have shown efficacy and safety of these drugs in elderly patients, without a significant increase of cardiac adverse event rates despite the older age of subjects and the presence of several comorbidities [143]. However, HF can frequently occur in up to 25% of elderly patients (mean age, 77.3 years) with MDS after a mean of 7.4 cycles, and history of cardiac diseases, red blood cell transfusion dependency, and increased levels of WT-1 are proposed as risk factors for HF during demethylating agent treatment [143]. Sporadic cases of pericarditis and cardiomyopathy are reported [144,145].

## 6. Monoclonal Antibodies

Monoclonal antibodies (MoAbs) are designed to recognize and bind tumor-specific antigens and direct against malignant cells through three types of immune responses resulting in tumor cell killing: antibody-dependent cell-mediated cytotoxicity (ADCC), complement-dependent cytotoxicity (CDC), and antibody-dependent phagocytosis (ADCP) [146]. In addition, MoAbs can bind receptors or molecules and interfere with specific signaling pathways, such as checkpoint inhibitors that bind PD-1 and stop its inhibitory signals in T cells [147]. In hematology, physiological expression of surface and intracellular markers in hematopoietic stem and progenitor cell compartment and in mature cells is well known, as well as neoplastic cell immunophenotype, which is characterized by the aberrant expression of lineage-specific antigens [148]. Therefore, malignant cells can be easily identified and targeted by specific MoAbs. In B cell non-Hodgkin lymphomas (NHL), neoplastic cells frequently show positivity for the CD20 antigen, a proposed calcium channel, which expression is normally lost on mature B cells, plasma blasts, and plasma cells. MoAbs direct against CD20 have radically changed clinical outcomes of B cell NHL, as neoplastic cells can be specifically targeted by those MoAbs and killed through ADCC or CDC mechanisms. Rituximab is a murine–human chimeric first-generation anti-CD20 MoAb, while obinutuzumab and ofatumumab are second-generation humanized or fully human anti-CD20 MoAbs that are less immunogenic and more effective in inducing apoptosis in B cells compared to rituximab [149,150]. MoAbs and their cardiotoxic effects are summarized in Table 1, showing that infusion-related reactions, hypertension or hypotension, and arrhythmias are the most common cardiotoxicity.

MoAbs against CD38, a type II transmembrane glycoprotein with ectoenzymatic activities, are used in MM treatment because neoplastic plasma cells highly express this surface marker [151]. Daratumumab, a fully human MoAb, and isatuximab, a chimeric MoAb recently approved by the FDA, are the two anti-CD38 antibodies used in clinical practice for MM treatment in association with IMiDs, proteasome inhibitors, and steroids [152]. Other promising MoAbs in MM treatment are elotuzumab, an anti-SLAMF7 (or CD319), approved in combination with lenalidomide and dexamethasone in relapsed/refractory MM patients [153], and belantamab mafodotin, an anti-B cell maturating antigen (BCMA) MoAb conjugated with a monomethyl auristatin F, approved as monotherapy in MM patients with disease progression who are refractory to proteasome inhibitors, IMiDs, and anti-CD38 MoAbs [154]. Brentuximab vedotin is an anti-CD30 MoAb conjugated with auristatin E, an anti-microtubule agent, and is currently used in CD30^+^ lymphoproliferative neoplasmas, such as advanced stage or relapsed/refractory Hodgkin lymphoma (HL), systemic anaplastic large cell lymphoma (ALCL), and CD30^+^ cutaneous T cell lymphomas [155,156]. In addition, two checkpoint inhibitors, the anti-PD-1 nivolumab and pembrolizumab MoAbs, have been approved for classical HL in relapsed/refractory patients after autologous stem cell transplantation or treatment with brentuximab vedotin [157]. Two new MoAbs approved for treatment of relapsed or refractory diffuse large B cell lymphoma (DLBCL) are the anti-CD19 tafasitamab, and the anti-CD79a conjugated with monomethyl auristatin E polatuzumab vedotin [158,159]. The most frequent cardiotoxicity is infusion-related reactions, hypertension, and arrhythmias, such as AF or tachycardia [160]; however, the exact mechanisms of non-infusion-related cardiotoxicity are still under investigation. For checkpoint inhibitors, myocarditis and pericardial diseases can be caused by autoimmune reactions against cardiac tissue triggered by hyperactivation of immune cells [161], while conjugated MoAbs, such brentuximab vedotin, can directly induce cardiotoxicity through off-target effects of coupled cytotoxic agents, such as auristatin E, an anti-mitotic drug.

## 7. Conclusions

Drugs in onco-hematology have both therapeutic benefits and toxicities that can compromise clinical response to treatment, worse management, and compliance of patients, and can dramatically reduce quality of life because of serious and late- or long-term adverse events, such as cardiotoxicity [1]. Cardiovascular manifestations can vary in types and time of onset depending on presence of risk factors, gene polymorphisms, anti-cancer drug used and cumulative dose, and presence of pre-existing conditions. The task force for cancer treatments and cardiovascular toxicity of the European Society of Cardiology (ESC) has outlined practice guidelines for prevention and attenuation of cardiovascular complications of cancer therapy [1]. The first action that a medical doctor must take before initiation of anti-cancer drug treatment is identification of cardiovascular risk factors and pre-existing conditions that are risk factors of cardiotoxicity in patients treated with BTK or JAK inhibitors, dasatinib, bosutinib, or ponatinib [1,162]. Additional risk factors of QT prolongation, such as concomitant use of drugs that increase QT intervals, should be identified, as QT prolongation is one of the most common arrhythmias related to the use of BTK and PI3K inhibitors and nilotinib or bosutinib. Patients with pre-existing conditions or risk factors who must start a cardiotoxic therapy should receive an appropriate treatment for their cardiovascular disease using ACE-Is or ARBs, β-blockers, statins, and aerobic exercise [1]. In some cases, cardioprotective molecules can be employed, such as dexrazoxane or carvedilol, which can significantly reduce troponin levels and diastolic dysfunction [1,163,164]. Patients without baseline risk factors and normal LVEF can also benefit from primary preventive pharmacologic treatment [162]. Cumulative dose of known cardiotoxic drugs should be reduced as soon as possible, such as for nilotinib, ponatinib, or FLT3 inhibitors, or a single daily dose should be used for certain drugs, such as dasatinib, in order to reduce the risk of pleural and pericardial effusion [88]. Therefore, cardiotoxicity prevention can be achieved by treating pre-existing conditions, reducing cardiovascular risk, and using cardioprotective agents, while early detection of cardiotoxicity can be realized with a close monitoring of patients with echocardiographic evaluation of cardiac functions and peripheral system, and with blood tests, such as troponin or BNP levels [9]. An even closer monitoring could be performed in old and frail patients who receive BTK inhibitors, bosutinib, ponatinib, IMiDs, or bortezomib, and in those who have long-course treatment with BTK and BCR/ABL inhibitors or bortezomib. Patients could be monitored for cardiotoxicity also after the end of treatment, such as for ponatinib-treated patients who can develop cardiac adverse events after several months of drug discontinuation [103,104].

In conclusion, cardiotoxicity is an increasing and complex problem as novel targeted therapies have several off-target effects negatively influencing cardiovascular functions and structure, thus worsening quality of life of long-term survivors. In addition, literature on cardiotoxicity is constantly changing as follow-up increases for novel drugs and late- and long-term manifestations occur. Hematologists should closely work with specialized cardiologists for a better clinical management of long-term survivors, and also of elderly and frail patients with MM or AML who already have an increased risk of cardiovascular events because of their hematological condition and the presence of several comorbidities.

## Figures and Tables

**Figure 1 life-10-00344-f001:**
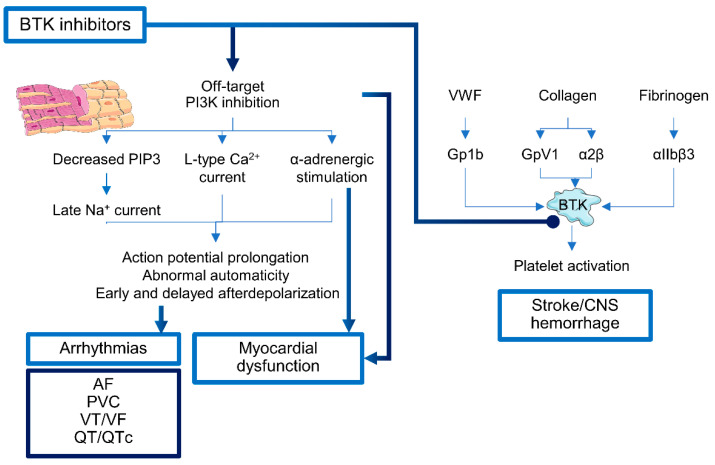
Cardiotoxicity of Bruton’s tyrosine kinase (BTK) inhibitors. BTK inhibitors can interfere with phosphoinositide 3-kinase (PI3K) pathways in cardiomyocytes, influence normal ion current, and cause action potential prolongation and abnormal automaticity leading to arrhythmias, such as atrial fibrillation (AF), premature ventricular contractions (PVC), ventricular tachycardia or fibrillation (VT/VF), and QT interval prolongation (QT/QTc). In platelets, BTK is activated by Von-Willebrand factor (VWF), collagen, and fibrinogen binding to cognate glycoproteins (e.g., Gp1b), leading to platelet activation. BTK inhibition in platelets is associated to central nervous system (CNS) bleeding or ischemia.

**Figure 2 life-10-00344-f002:**
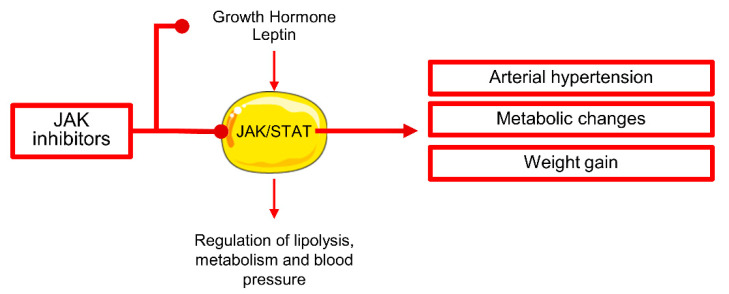
Cardiotoxicity of Janus kinase (JAK) inhibitors. JAK inhibitors can interfere with growth hormone and leptin pathways in adipocytes, causing dysregulation of lipolysis, metabolism, and blood pressure homeostasis. In adipocytes, JAK/signal transducer and activator of transcription (STAT) pathway, mainly through STAT5, is involved in regulation of lipolysis, glucose and lipid metabolism, response to insulin stimulation, and regulation of blood pressure. Inhibition of JAK/STAT in adipocytes can contribute to metabolic changes leading to weight gain and arterial hypertension development.

**Figure 3 life-10-00344-f003:**
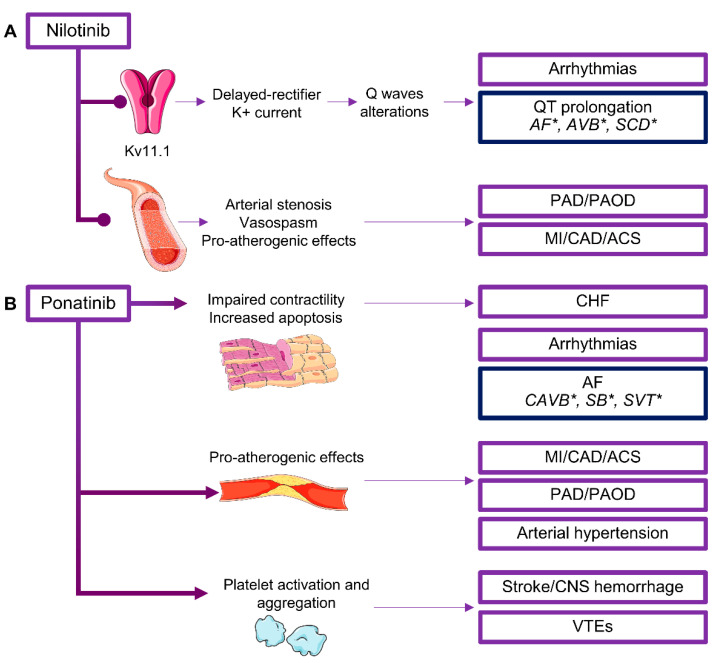
Cardiotoxicity of B cell receptor (BCR)/ABL tyrosine kinase inhibitors (TKIs). Cardiotoxicity of (**A**) nilotinib and (**B**) ponatinib. Inhibitors of BCR/ABL activity cause various forms of cardiotoxicity, such as congestive heart failure (CHF), myocardial infarction (MI), acute coronary syndrome (ACS), coronary artery disease (CAD), peripheral artery disease (PAD), peripheral arterial occlusive disease (PAOD), venous thromboembolisms (VTEs), and arrhythmias. SCD, sudden cardiac death; SVT, supraventricular tachycardia; CAVB, complete atrioventricular block; SB, sinus bradycardia. * Uncommon and rare cardiotoxicity (frequency <1% or case reports).

**Table 1 life-10-00344-t001:** Cardiotoxicity of monoclonal antibodies (MoAbs) in hematology.

MoAb	Target	Indications	Cardiotoxicity
Rituximab	CD20	NHL CLL	IRRs
			Hypertension (6–12%)
			Transient hypotension (10%)
			SVT, AF
			Takotsubo *
			CAVB *, ST *, SB *, PVC *, VT *, QTc/TdP *, SCD *
Obinutuzumab	CD20	CLL/FL	IRRs, SCD *, HF *
Ofatumumab	CD20	CLL	IRRs
			Hypertension/hypotension
			Tachycardia
			AF *, HF *, MI *, pericarditis *
Daratumumab	CD38	MM	IRRs
			Hypertension
			AF
			SCD *
Isatuximab	CD38	MM	IRRs
			AF
Elotuzumab	SLAMF7	MM	IRRs
			DVT
Brentuximab vedotin	CD30	HL	ST (6%)
			Pericardial effusion (3%)
			CHF *, MI *
Nivolumab	PD-1	HL	Myocarditis
			Pericardial diseases
			Stress cardiomyopathy
			VT
			CAVB *, SCD *
Pembrolizumab	PD-1	HL	Myocarditis
			Pericardial diseases
			Stress cardiomyopathy
			ST *, AF *, PVC *, VT *, SCD *
Gemtuzumab ozogamicin	CD33	AML	VOD (2%)
			Tachycardia, ST, SVT (13%)
			Hypertension (17.3%)
Blinatumomab	CD19/CD3	R/R Ph-ALL	Tachycardia
			HF *
Belantamab mafodotin	BCMA	MM	Not reported
Inotuzumab ozogamicin	CD22	Ph+ ALL	VOD
			QT/QTc
Moxetumomab pasudotox	CD22	R/R HCL	HUS *
			Pericardial/pleural effusion *
			Hypertension/hypotension *
			Tachycardia *
Tafasitamab	CD19	DLBCL	Pulmonary embolism (4%)
			AF (2%)
			CHF (2%)
Polatuzumab vedotin	CD79a	DLBCL	Hypotension
Ravulizumab	C5a	PNH HUS	Hypertension/hypotension *
Eculizumab	C5a	PNH HUS	Hypertension
			Tachycardia/Palpitation
			Cardiomyopathy *
			Hypotension *
Emapalumab	IFNγ	R/R HLH	Hypertension (41%)
			Tachycardia (12%)
			Bradycardia *
Siltuximab	IL-6	Castleman disease	Hypertension
			Peripheral edema (26%)

Abbreviations. MoAb, monoclonal antibodies; NHL, non-Hodgkin lymphomas; CLL, chronic lymphocytic leukemia; IRRs, infusion-related reactions; SVT, supraventricular tachycardia; AF, atrial fibrillation; CAVB, complete atrioventricular block; ST, sinus tachycardia; SB, sinus bradycardia; PVC, premature ventricular contractions; VT, ventricular tachycardia; QT/QTc, QT interval prolongation; TdP, torsade de pointes; SCD, sudden cardiac death; FL, follicular lymphoma; HF, heart failure; MI, myocardial infarction; MM, multiple myeloma; DVT, deep vein thrombosis; HL, Hodgkin lymphoma; CHF, congestive heart failure; PD-1, programmed cell death protein 1; AML, acute myeloid leukemia; VOD, veno-occlusive disease; R/R, relapsed/refractory; Ph, Philadelphia chromosome; ALL, acute lymphoblastic leukemia; BCMA, B cell maturation antigen; HCL, hairy cell leukemia; HUS, hemolytic uremic syndrome; DLBCL, diffuse large B cell lymphoma; PNH, paroxysmal nocturnal hemoglobinuria; IFNγ, interferon gamma; HLH, hemophagocytic lymphohistiocytosis; IL-6, interleukin-6. * *Uncommon and rare cardiotoxicity (frequency <1% or case reports)*.

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
