# Peer review of "Cardiotoxicity of Novel Targeted Hematological Therapies"

_life, 2020, doi:10.3390/life10120344_

Round 1

Reviewer 1 Report

The work Cardiotoxicity of novel targeted hematological therapies is very interesting, up to date and with value. Some issues may improve its impact.

- the abstract is not focused on theme while targeted therapies used in hematology are mentioned in the bottom.

- although I agree that old chemotherapy is better studied, concerning e.g. cytarabine studies are lacking, possibly due to poor survival rates, but should be mentioned.

  • Change legend of figure 1 (major cardiotoxicies??)
  • Some parts are too descriptive namely page 3
  • Figures need improvement: naïve and not clear, they do not bring anything new. I suggest new ones related with mechanism

  • Clarify this sentence: ‘IDH-DH can also occur during enasidenib treatment; however, no links between IDH-DS-related deaths and IDH inhibitor administration are documented [51-52].’
  • The most frequent cardiotoxicity and not carditoxicities
  • Less detailed on the studies pointed. More objectivity

  • There are 2 tables, both named Table 1. Moreover, they are not addressed in the text in the exhaustively way they should. Moreover, referring to the second table, what are the drugs that depend on those risk for cardiotoxicity development?

  • The conclusions are interesting, but as the major part of the paper refers to scientific and mechanistic approach, while the conclusions are basically medical and not follows the general features of the paper.

Author Response

The work Cardiotoxicity of novel targeted hematological therapies is very interesting, up to date and with value. Some issues may improve its impact.

Comment 1: the abstract is not focused on theme while targeted therapies used in hematology are mentioned in the bottom.

Response to Comment 1: We thank the Reviewer for this comment and the following text was added to the abstract “For example, Bruton’s tyrosine kinase (BTK) inhibitors interfere with class I Phosphoinositide 3-kinase isoforms involved in cardiac hypertrophy, contractility, and in regulation of various channel forming proteins; thus off-target effects of BTK inhibitors are associated with increased frequency of arrhythmias, such as atrial fibrillation, compared to standard chemotherapy.”

Comment 2: although I agree that old chemotherapy is better studied, concerning e.g. cytarabine studies are lacking, possibly due to poor survival rates, but should be mentioned.

Response to Comment 2: We thank the Reviewer for this helpful comment and we have added the following text from line 52 to 62: “Chemotherapy is employed for treatment of Hodgkin and non-Hodgkin lymphomas, myeloid and lymphoid leukemias, and before hematopoietic stem cell transplantation as conditioning regimens. Chemotherapy-related cardiac complications and mortality have been historically reported at high incidence rates in earlier studies; however, optimization of cumulative dosage, such as introduction of reduced-intensity conditioning regimens, or different chemotherapeutic scheme, such as from combination of vincristine, methotrexate, cyclophosphamide and prednisone (MOMP) to adriamycin, bleomycin, vinblastine and dacarbazine (ABVD) for treatment of Hodgkin lymphoma, has significantly reduced the impact of cardiotoxicity in hematological patients [10-11]. In last decades, standard chemotherapy has been also associated or replaced with novel targeted therapies which are used for rescue refractory/relapsed patients.”

Comment 3: Change legend of figure 1 (major cardiotoxicies??)

Response to Comment 3: Figure legend 1 now reads “Types of cardiotoxicity”.

Comment 4: Some parts are too descriptive namely page 3.

Response to Comment 4: We thank the Reviewer for the comment, and we have shortened all sections accordingly.

Comment 5: Figures need improvement: naïve and not clear, they do not bring anything new. I suggest new ones related with mechanism.

Response to Comment 5: Figures 1, 4, and 5 have been removed. We have changed remaining figures by adding biological mechanisms of cardiotoxicity.

Comment 6: Clarify this sentence: ‘IDH-DH can also occur during enasidenib treatment; however, no links between IDH-DS-related deaths and IDH inhibitor administration are documented [51-52].’

Response to Comment 6: On lines 182-184, the sentence now reads “Recently, a case of myopericarditis and cardiogenic shock following an IDH inhibitor-induced differentiation syndrome (IDH-DS) has been reported during enasidenib treatment [50-52].”

Comment 7: The most frequent cardiotoxicity and not carditoxicities.

Response to Comment 7: We thank the Reviewer for the comment. We have changed the term throughout the text.

Comment 8: Less detailed on the studies pointed. More objectivity,

Response to Comment 8: Please refer to Response to Comment 4.

Comment 9: There are 2 tables, both named Table 1. Moreover, they are not addressed in the text in the exhaustively way they should. Moreover, referring to the second table, what are the drugs that depend on those risk for cardiotoxicity development?

Response to Comment 9: Table 2 has been removed. For Table 1, on lines 446-448, the following text was added “MoAbs and their cardiotoxic effects are summarized in Table 1 showing that infusion-related reactions, hypertension or hypotension, and arrhythmias are the most common cardiotoxicity.”

Comment 10: The conclusions are interesting, but as the major part of the paper refers to scientific and mechanistic approach, while the conclusions are basically medical and not follows the general features of the paper.

Response to Comment 10: We thank the Reviewer for the comment. We have changed the Conclusions section as follows “Drugs in onco-hematology have both therapeutic benefits and toxicities which can compromise clinical response to treatment, worse management and compliance of patients, and can dramatically reduce quality of life because of serious and late- or long-term adverse events, such as cardiotoxicity [1]. Cardiovascular manifestations can vary in types and time of onset depending on presence of risk factors, gene polymorphisms, anti-cancer drug used and cumulative dose, and presence of pre-existing conditions. The Task Force for cancer treatments and cardiovascular toxicity of the European Society of Cardiology (ESC) has outlined practice guidelines for prevention and attenuation of cardiovascular complications of cancer therapy [1]. The first action that a medical doctor must take before initiation of anti-cancer drug treatment is identification of cardiovascular risk factors and pre-existing conditions which are risk factors of cardiotoxicity in patients treated with BTK or JAK inhibitors, dasatinib, bosutinib, or ponatinib [1,163]. Additional risk factors of QT prolongation, such as concomitant use of drugs that increase QT intervals, should be identified as QT prolongation is one of the most common arrhythmias related to the use of BTK and PI3K inhibitors and nilotinib or bosutinib. Patients with pre-existing conditions or risk factors who must start a cardiotoxic therapy should receive an appropriate treatment for their cardiovascular disease using ACE-Is o ARBs, β-blockers, statins, and aerobic exercise [1]. In some cases, cardioprotective molecules can be employed, such as dexrazoxane or carvedilol that can significantly reduce troponin levels and diastolic dysfunction [1,164-165]. Patients without baseline risk factors and normal LVEF can also benefit from primary preventive pharmacologic treatment [163]. Cumulative dose of known cardiotoxic drugs should be reduced as soon as possible, such as for nilotinib, ponatinib, or FLT3 inhibitors; or prefer a single daily dose for certain drugs, such as dasatinib, to reduce the risk of pleural and pericardial effusion [88]. Therefore, cardiotoxicity prevention can be achieved by treating pre-existing conditions, reducing cardiovascular risk, and using cardioprotective agents; while early detection of cardiotoxicity can be realized with a close monitoring of patients with echocardiographic evaluation of cardiac functions and peripheral system, and with blood tests, such as troponin or BNP levels [9]. An even closer monitoring could be performed in old and frail patients who receive BTK inhibitors, bosutinib, ponatinib, IMiDs, or bortezomib, and in those who have long-course treatment with BTK and BCR/ABL inhibitors or bortezomib. Patients could be monitored for cardiotoxicity also after the end of treatment, such as for ponatinib-treated patients who can develop cardiac adverse events after several months of drug discontinuation [103-104].

In conclusion, cardiotoxicity is an increasing and complex problem as novel targeted therapies have several off-target effects negatively influencing cardiovascular functions and structure thus worsening quality of life of long-term survivors. In addition, literature on cardiotoxicity is constantly changing as follow-up increases for novel drugs and late- and long-term manifestations occur. Hematologists should closely work with specialized cardiologists for a better clinical management of long-term survivors, and also of elderly and frail patients with MM or AML who already have an increased risk of cardiovascular events because of their hematological condition and the presence of several comorbidities.”

Reviewer 2 Report

In the present review (Cardiotoxicity of novel targeted hematological therapies), authors expose an interesting actualization in potential cardiotoxic effects of onco-hematological chemotherapies. They review the most used drugs for treatment of oncological disorders with their most common cardiac side effects.

This review is truly complete with an appropriate and updated bibliography. And, although it is globally well conducted, there are several concerns that, if addressed, may improve it for further publication:

Concerns:

1) In the introduction authors highlight the lack of biomarkers of cardiotoxicity in clinical practice [line 40]. But there are several publications addressing the role of different biomarkers (mainly natriuretic peptides and troponin) as early predictors of chemotherapy cardiac side effects and its predictive value. For example it has been recently published a metanalysis with the role of both, troponin and brain natriuretic peptides, as predictors of cardiotoxicity (1).

2) Authors should broaden the information about cardiotoxicity prevention and early diagnosis.

3) Betablockers are probably the most useful drugs for cardiotoxicity prevention. In this sense authors should include evidence in favor of these treatments, not just the guidelines recommendations. For example, it has been recently published the role of Carvedilol for cardiotoxicity prevention (2).

Author Response

In the present review (Cardiotoxicity of novel targeted hematological therapies), authors expose an interesting actualization in potential cardiotoxic effects of onco-hematological chemotherapies. They review the most used drugs for treatment of oncological disorders with their most common cardiac side effects.

This review is truly complete with an appropriate and updated bibliography. And, although it is globally well conducted, there are several concerns that, if addressed, may improve it for further publication:

Concerns:

Comment 1: In the introduction authors highlight the lack of biomarkers of cardiotoxicity in clinical practice [line 40]. But there are several publications addressing the role of different biomarkers (mainly natriuretic peptides and troponin) as early predictors of chemotherapy cardiac side effects and its predictive value. For example it has been recently published a metanalysis with the role of both, troponin and brain natriuretic peptides, as predictors of cardiotoxicity (1).

Response to Comment 1: We thank the Reviewer for this suggestion. On lines 47-49, the following text was added: “Prolonged augmented circulating troponin levels might also help in early detection of LV dysfunction during chemotherapy treatment, while (N-terminal pro) brain natriuretic peptide (BNP/NT-proBNP) levels can increase after therapy [9].

Comment 2: Authors should broaden the information about cardiotoxicity prevention and early diagnosis.

Comment 3: Betablockers are probably the most useful drugs for cardiotoxicity prevention. In this sense authors should include evidence in favor of these treatments, not just the guidelines recommendations. For example, it has been recently published the role of Carvedilol for cardiotoxicity prevention (2).

Response to Comments 2 and 3: We have added this information in the Conclusions section that now reads “Drugs in onco-hematology have both therapeutic benefits and toxicities which can compromise clinical response to treatment, worse management and compliance of patients, and can dramatically reduce quality of life because of serious and late- or long-term adverse events, such as cardiotoxicity [1]. Cardiovascular manifestations can vary in types and time of onset depending on presence of risk factors, gene polymorphisms, anti-cancer drug used and cumulative dose, and presence of pre-existing conditions. The Task Force for cancer treatments and cardiovascular toxicity of the European Society of Cardiology (ESC) has outlined practice guidelines for prevention and attenuation of cardiovascular complications of cancer therapy [1]. The first action that a medical doctor must take before initiation of anti-cancer drug treatment is identification of cardiovascular risk factors and pre-existing conditions which are risk factors of cardiotoxicity in patients treated with BTK or JAK inhibitors, dasatinib, bosutinib, or ponatinib [1,163]. Additional risk factors of QT prolongation, such as concomitant use of drugs that increase QT intervals, should be identified as QT prolongation is one of the most common arrhythmias related to the use of BTK and PI3K inhibitors and nilotinib or bosutinib. Patients with pre-existing conditions or risk factors who must start a cardiotoxic therapy should receive an appropriate treatment for their cardiovascular disease using ACE-Is o ARBs, β-blockers, statins, and aerobic exercise [1]. In some cases, cardioprotective molecules can be employed, such as dexrazoxane or carvedilol that can significantly reduce troponin levels and diastolic dysfunction [1,164-165]. Patients without baseline risk factors and normal LVEF can also benefit from primary preventive pharmacologic treatment [163]. Cumulative dose of known cardiotoxic drugs should be reduced as soon as possible, such as for nilotinib, ponatinib, or FLT3 inhibitors; or prefer a single daily dose for certain drugs, such as dasatinib, to reduce the risk of pleural and pericardial effusion [88]. Therefore, cardiotoxicity prevention can be achieved by treating pre-existing conditions, reducing cardiovascular risk, and using cardioprotective agents; while early detection of cardiotoxicity can be realized with a close monitoring of patients with echocardiographic evaluation of cardiac functions and peripheral system, and with blood tests, such as troponin or BNP levels [9]. An even closer monitoring could be performed in old and frail patients who receive BTK inhibitors, bosutinib, ponatinib, IMiDs, or bortezomib, and in those who have long-course treatment with BTK and BCR/ABL inhibitors or bortezomib. Patients could be monitored for cardiotoxicity also after the end of treatment, such as for ponatinib-treated patients who can develop cardiac adverse events after several months of drug discontinuation [103-104].

In conclusion, cardiotoxicity is an increasing and complex problem as novel targeted therapies have several off-target effects negatively influencing cardiovascular functions and structure thus worsening quality of life of long-term survivors. In addition, literature on cardiotoxicity is constantly changing as follow-up increases for novel drugs and late- and long-term manifestations occur. Hematologists should closely work with specialized cardiologists for a better clinical management of long-term survivors, and also of elderly and frail patients with MM or AML who already have an increased risk of cardiovascular events because of their hematological condition and the presence of several comorbidities.”

Reviewer 3 Report

In this review article, Giudice and colleagues summarize the most relevant cardiotoxic effects associated with the administration of novel targeted haematological therapies. Authors report that although these drugs significantly increased the life expectancy of patients, their use is responsible for the onset of short- and long-term cardiotoxicity. Unlike the classical approaches with radiotherapy and chemotherapy, target therapies negatively affect cardiac function by interfering with specific pathways involved in the homeostasis of the cardiovascular system. Additionally, while the adverse cardiac effects of the first-generation agents, such as tyrosine kinase inhibitors (TKIs), are well described, less is known about the recently approved drugs. Authors claim that several factors contribute to the clinical limitations of these therapeutic strategies, including aging, previous/concomitant treatments, comorbidities and availability of long-term follow-ups. Concerning all these points, a better understanding of the factors and the mechanisms involved in target therapy-induced cardiotoxicity can help haematologists to identify the best therapeutic option.

I appreciate authors’ effort of summarizing a huge amount of clinical evidence. However, the quality of the work could be significantly improved by addressing the following points.

MAJOR POINTS:

-The cross inhibition of PI3Ks, mediated by Bruton’s tyrosine kinase (BTK) inhibitors in the insurgence of atrial fibrillation, needs to be described in detail because of the complexity of the PI3K pathway. Indeed, while it has been reported that PI3Kα/δ/β have an anti-arrhythmic effect (PMID:2600602), PI3Kγ has been shown to have a detrimental role in the cardiac response to chemotherapy treatment (PMID:  29348263). Additionally, authors could discuss putative therapeutic strategies aimed at restoring the activity of the PI3K/Akt pathway (PMID:25640634).

-The approval of the JAK2 inhibitor Fedratinib by FDA includes a “black box warning” that indicates the risk of fatal encephalopathy (PMID:32343799). Authors might discuss the ongoing development of new and safety strategies, including Momelotinib and Pacritinib (PMID: 31511492)

-In the “BCR/ABL inhibitors” section, the impact of BCR/ABL on cardiomyocytes needs to be revised. First, there is no BCR/ABL blockade in cardiomyocytes and the cardiotoxicity is induced by the off-target effects of BCR/ABL inhibitors. Second, cardiomyocytes are known to be terminally differentiated cells and it is not clear how FGFR inhibition causes modifications in proliferation and differentiation of this type of cell (Pag.7, lanes 276-277).

-The quality of the writing could be improved, including the fluency of the discussion. In some points, authors’ arguments are difficult to follow because data are not linked in a logic way.  

MINOR POINTS

-  Authors could indicate the type of mutation occurring in the calreticulin gene, as previously done for JAK2 and MPL.

-I strongly suggest making abbreviations consistent (i.e. “BCR/ABL” or “BCL-ABL”). 

Author Response

In this review article, Giudice and colleagues summarize the most relevant cardiotoxic effects associated with the administration of novel targeted haematological therapies. Authors report that although these drugs significantly increased the life expectancy of patients, their use is responsible for the onset of short- and long-term cardiotoxicity. Unlike the classical approaches with radiotherapy and chemotherapy, target therapies negatively affect cardiac function by interfering with specific pathways involved in the homeostasis of the cardiovascular system. Additionally, while the adverse cardiac effects of the first-generation agents, such as tyrosine kinase inhibitors (TKIs), are well described, less is known about the recently approved drugs. Authors claim that several factors contribute to the clinical limitations of these therapeutic strategies, including aging, previous/concomitant treatments, comorbidities and availability of long-term follow-ups. Concerning all these points, a better understanding of the factors and the mechanisms involved in target therapy-induced cardiotoxicity can help haematologists to identify the best therapeutic option.

I appreciate authors’ effort of summarizing a huge amount of clinical evidence. However, the quality of the work could be significantly improved by addressing the following points.

MAJOR POINTS:

Comment 1: The cross inhibition of PI3Ks, mediated by Bruton’s tyrosine kinase (BTK) inhibitors in the insurgence of atrial fibrillation, needs to be described in detail because of the complexity of the PI3K pathway. Indeed, while it has been reported that PI3Kα/δ/β have an anti-arrhythmic effect (PMID: 2600602), PI3Kγ has been shown to have a detrimental role in the cardiac response to chemotherapy treatment (PMID: 29348263). Additionally, authors could discuss putative therapeutic strategies aimed at restoring the activity of the PI3K/Akt pathway (PMID: 25640634).

Response to Comment 1: We thank the Reviewer for this helpful comment, and we hope we have satisfactorily discussed this point. On lines 99-117, the following text was added “BTK inhibitors can interfere with all Class I PI3K isoforms (PI3Kα, PI3Kα, PI3Kγ, and PI3Kδ) which are differently expressed in various tissues and involved in cardiac functions. Class I PI3Kα has an essential role in physiological cardiac hypertrophy and contractility, and is activated in cardiomyocytes by insulin or insulin-like growth factor-1 (IGF-1), while in endothelial cells, fibroblasts and vascular smooth muscle cells by fibroblast growth factor (FGF), platelet-derived (PDGF), and vascular endothelial growth factor (VEGF). In mouse models, PI3Kα suppression worsens hypertrophic cardiomyopathy caused by pressure overload or myocardial infarction (MI), while PI3Kα activation ameliorates hypertrophic and dilated cardiomyopathy [27-28]. PI3Kα is also involved in regulation of various channel forming proteins, such as K+: Kir, Ca2+: Cav1; or Na+: SCN5A; direct inhibition of PI3Kα or at the receptor level (e.g. by ibrutinib) causes activation of late Na current (INa-L) through PIP3 reduction resulting in enhanced action potential and QT prolongation. In addition, PI3Kα inhibition affects L-type Ca2+ current (ICa,L), and modulates Ca2+ cycling and α-adrenergic stimulation favoring action potential prolongation, abnormal automaticity, and early and delayed afterdepolarization [27]. While, PI3Kγ, mainly expressed in leukocytes and cardiac cells, is upregulated during atherosclerosis, has opposite inotropic functions compared to PI3Kα, and decreases myocardial β-adrenergic receptor (β-AR) under stress conditions, such as during congestive heart failure (CHF) [27,29]. Therefore, the use of agents that block the activation of INa-L (e.g., ranolazine) or upregulate ion channel expression might reduce the impact of PI3K inhibition-related cardiotoxicity [27,30].”

Comment 2: The approval of the JAK2 inhibitor Fedratinib by FDA includes a “black box warning” that indicates the risk of fatal encephalopathy (PMID:32343799). Authors might discuss the ongoing development of new and safety strategies, including Momelotinib and Pacritinib (PMID: 31511492).

Response to Comment 2: We thank the Reviewer for the comment. On lines 217-220, the following text was added “However, fedratinib has received a “black-box warning” because of the increased risk of fatal encephalopathy including Wernicke encephalopathy [71]. Other JAK inhibitors under investigation are momelotinib and pacritinib, a dual JAK/FLT3 inhibitor, that cause less hematological adverse events and cardiotoxicity [72].

Comment 3: In the “BCR/ABL inhibitors” section, the impact of BCR/ABL on cardiomyocytes needs to be revised. First, there is no BCR/ABL blockade in cardiomyocytes and the cardiotoxicity is induced by the off-target effects of BCR/ABL inhibitors. Second, cardiomyocytes are known to be terminally differentiated cells and it is not clear how FGFR inhibition causes modifications in proliferation and differentiation of this type of cell (Pag.7, lanes 276-277).

Response to Comment 3: We apologize for misleading sentences. On lines 286-290, we have rephrased them as follows “Mechanisms of cardiotoxicity are still under investigation; however, off-target effects especially on PI3K and Akt signaling pathways might be responsible of cardiotoxicity [104]. Off-target FGFR inhibition causes modifications in vitro proliferation and differentiation of cardiomyocytes; FLT3 and c-Jun blockade is related to apoptosis; while PDGFR, VEGFR, and c-Src inhibition induces contractile alterations [104].”

Comment 4: The quality of the writing could be improved, including the fluency of the discussion. In some points, authors’ arguments are difficult to follow because data are not linked in a logic way. 

Response to Comment 4: We have checked the manuscript and the discussion. We hope we have made it more fluent and linked in a logical way.

MINOR POINTS

Comment 5: Authors could indicate the type of mutation occurring in the calreticulin gene, as previously done for JAK2 and MPL.

Response to Comment 5: On lines 194-195, missing information has been added as follows “and mutations in exon 9 of the calreticulin (CALR) gene as a 52-bp deletion (L367fs*46) or a 5-bp insertion (K385fs*47) [57].”

Comment 6: I strongly suggest making abbreviations consistent (i.e. “BCR/ABL” or “BCL-ABL”).

Response to Comment 6: We apologize for inconsistency. We have changed BCR-ABL to BCR/ABL, and we have carefully checked the manuscript for other inconsistencies.

Round 2

Reviewer 1 Report

Figure 1 is much better, nevertheless, if authors pretend to maintain % of AF, a reference in the legend must be added.

Figure 2 legend needs improvement and if % is maintained a reference needs adding.

Some typos: like line 288 and the lack of space

Figure 3 (see comments before); define also the significance of uncommon? How often?

Author Response

Comment 1: Figure 1 is much better, nevertheless, if authors pretend to maintain % of AF, a reference in the legend must be added.

Response to Comment 1: Percentages were removed from all figures.

Comment 2: Figure 2 legend needs improvement and if % is maintained a reference needs adding.

Response to Comment 2: We thank the Reviewer for the comment. Figure 2 caption now reads “Figure 2. Cardiotoxicity of JAK inhibitors. JAK inhibitors can interfere with growth hormone and leptin pathways in adipocytes causing dysregulation of lipolysis, metabolism, and blood pressure homeostasis. In adipocytes, JAK/STAT pathway mainly through STAT5 is involved in regulation of lipolysis, glucose and lipid metabolism, response to insulin stimulation, and regulation of blood pressure. Inhibition of JAK/STAT in adipocytes can contribute to metabolic changes leading to weight gain and arterial hypertension development.”

Comment 3: Some typos: like line 288 and the lack of space.

Response to Comment 3: We apologize for typos and we have corrected as indicated.

Comment 4: Figure 3 (see comments before); define also the significance of uncommon? How often?

Response to Comment 4: Percentages were removed as suggested. The following text was added to figure legends and Table 1 caption “*Uncommon and rare cardiotoxicity (frequency <1% or case reports).

Reviewer 3 Report

Authors satisfactorily addressed all the points that I have raised. 

Author Response

We thank the Reviewer for the helpfull comments that have significantly improved the quality of our manuscript.